# Molecular Cloning and Characterization of *SQUAMOSA*-Promoter Binding Protein-Like Gene *FvSPL10* from Woodland Strawberry (*Fragaria vesca*)

**DOI:** 10.3390/plants8090342

**Published:** 2019-09-11

**Authors:** Jinsong Xiong, Yibo Bai, Chuangju Ma, Hongyu Zhu, Dan Zheng, Zongming Cheng

**Affiliations:** 1State Key Laboratory of Crop Genetics and Germplasm Enhancement, College of Horticulture, Nanjing Agricultural University, Nanjing 210095, China; 2Department of Plant Science, University of Tennessee, Knoxville, TN 37996, USA

**Keywords:** molecular cloning, characterization, *FvSPL10*, woodland strawberry

## Abstract

*SQUAMOSA*-promoter binding protein-like (SPL) proteins are plant-specific transcript factors that play essential roles in plant growth and development. Although many *SPL* genes have been well characterized in model plants like *Arabidopsis*, rice and tomato, the functions of *SPLs* in strawberry are still largely elusive. In the present study, we cloned and characterized *FvSPL10*, the ortholog of *AtSPL9*, from woodland strawberry. Subcellular localization shows *FvSPL10* localizes in the cell nucleus. The luciferase system assay indicates *FvSPL10* is a transcriptional activator, and both in vitro and in vivo assays indicate *FvSPL10* could bind to the promoter of *FvAP1* and activate its expression. Ectopic expression of *FvSPL10* in *Arabidopsis* promotes early flowering and increases organs size. These results demonstrate the multiple regulatory roles of *FvSPL10* in plant growth and development and lay a foundation for investigating the biological functions of *FvSPL10* in strawberry.

## 1. Introduction

As a class of development-related plant-specific transcription factor (TF), the most common feature of *SQUAMOSA*-promoter binding protein-like (SPL) TFs is that they all contain a highly conserved DNA binding domain. This is known as the *SQUAMOSA*-promoter-binding (SBP) domain. The SBP consists of two Zn-finger-like structures and one nuclear localization signal motif, which is approximately 76 amino acids in length [1]. Since the first identification in *Antirrhinum majus* [2], *SPL* genes have been discovered and cloned in almost all green plant species [3]. However, different from other plant-specific TFs, such as AP2/ERE (Apetala2/ethylene response factor) and NAC (NAM, ATAF and CUC) which contain more than one hundred members in their families, the *SPL* gene family in plant is relatively smaller. For example, the *SPL* family in maize, rice, *Arabidopsis*, tomato, grape and woodland strawberry contains only 31, 19, 16, 15, 18 and 14 members, respectively [4,5,6,7,8].

Though *SPLs* have been identified in various plants, only a few have been characterized. Additionally, of the *SPLs* that have been characterized, most are limited to model plants such as *Arabidopsis*, rice and tomato. In *Arabidopsis*, the 16 SPL TFs could be grouped into two classes according to their protein size. Class I includes five members: AtSPL1, AtSPL7, AtSPL12, AtSPL14 and AtSPL16. These are all large proteins that contain more than 800 amino acids. In contrast, the remaining 11 SPL TFs in class II are small proteins that contain less than 400 residues [1,9]. Studies have indicated that SPL TFs play important roles in plant growth and development. *AtSPL1* and *AtSPL12* regulate plant thermotolerance at the reproductive stage [10]. *AtSPL2*, *AtSPL10* and *AtSPL11* are functionally redundant genes that play roles in controlling the development of lateral organs at the reproductive phase [11]. *AtSPL3*, *AtSPL4* and *AtSPL5* are also functionally redundant genes that regulate floral meristem identity [12,13,14]. Interestingly, and in contrast to most of the other *SPLs* in *Arabidopsis* that mainly play roles in plant growth and development, *AtSPL6* was found to positively regulate plant innate immunity in defense against pathogens [15]. *AtSPL7* was found to help regulate Cu homeostasis [16,17], while *AtSPL8* helps to regulate pollen sac development [18]. *AtSPL9* and *AtSPL15* are a pair of functionally redundant genes that control plant juvenile-to-adult growth phase transition [19]. Moreover, *AtSPL9* was found to play roles in negatively regulating anthocyanin biosynthesis [20] and axillary bud formation [21], and *AtSPL9* and *AtSPL13* regulate trichome distribution [22]. Finally, *AtSPL14* functions in plant development and sensitivity to fungal toxin fumonisin B1 [23], while *AtSPL16* is a non-functional duplication of *AtSPL14* [3]. *OsSPL14* and *OsSPL16* are two well-characterized *SPL* genes in rice, the former was found to regulate plant architecture [24,25] and the latter regulate grain size [26]. In tomato, the most functional characterized *SPL* gene was *CNR* (*colorless non-ripening*), which has been identified as a key regulator involved in tomato fruit ripening [27]. In summary, as a class of plant-specific TF, most of the SPLs play critical roles in plant growth and development, especially in juvenile-to-adult transition through regulating flowering related genes. 

Strawberry is an economically important and nutritionally rich fruit crop. The market value of strawberry is not only determined by its quality, but also by its launch time. To fruit, the earlier it is put on the market, the more expensive it will be to purchase. However, fruit maturation is closely related to the flowering time, promoting fruit plant flowering could consequently lead to earlier fruit maturation. Therefore, promoting flowering is an important breeding goal for strawberry and other fruit crops. Although numerous studies in other plants have elucidated the important regulating roles of *SPL* genes in the plant phase transition, the function of strawberry *SPL* genes remains elusive. 

In a previous study, we systematically identified 14 *SPLs* in woodland strawberry [8]. Further tissue expression analysis indicated that one *FvSPL* gene, *FvSPL10* (*mrna22660.1*)*,* was mainly expressed in vegetative tissues, especially in the runner [8]. Because the expression pattern of *FvSPL10* is quite different from its *Arabidopsis* ortholog *AtSPL9*, and considering the important roles of *AtSPL9* during plant development in *Arabidopsis*, we decided to perform a detailed investigation about *FvSPL10* in this study. The results indicate that *FvSPL10* is a transcriptional activator that could specifically recognize and bind to the GTAC motif in the promoter of *FvAP1* and activate its expression. Ectopic expression of *FvSPL10* in *Arabidopsis* induces the expression of several flowering related genes, and therefore promotes early plant flowering (3–5 days). In addition, overexpression of *FvSPL10* in *Arabidopsis* increases flower and seed size. These results demonstrate the multiple regulatory roles of *FvSPL10* in plant growth and development, which provides valuable information for further investigating the function of *FvSPL10* on plant architecture and flowering-related traits in strawberry. 

## 2. Results

### 2.1. Cloning and Characterization of FvSPL10 in Woodland Strawberry

In our previous study, we identified 14 *SPL* genes in woodland strawberry and named *FvSPL1* to *FvSPL14* based on their chromosome location. Tissue expression analysis indicated *FvSPL10* was highly expressed in runners. The coding sequence (CDS) of *FvSPL10* is 1143 bp in length, which encodes 380 amino acids, with a predicted molecular mass of 40.04 kDa and an isoelectric point (pI) of 9.38. Phylogenetic analysis with *AtSPLs* indicated that *FvSPL10* was closely related to *AtSPL9* (Figure 1A). Further amino acid sequence alignments indicated that two SPLs shared 36% similarity, and the most conserved region was the N-terminus, which contains the SBP domain (Figure 1B). 

### 2.2. Subcellular Localization and Transcriptional Activity of FvSPL10 

Usually, TFs are located in the nucleus to regulate the expression of their target genes. To validate the subcellular localization of *FvSPL10*, we transiently expressed GFP-tagged *FvSPL10* fusion protein in tobacco leaves with GFP alone as a control. As expected, the GFP-FvSPL10 fusion protein exclusively localized in the cell nucleus, and GFP was distributed both at the cytoplasm and nucleus (Figure 2A). As a class of transcriptional regulators, TFs usually activate or repress the expression of their target genes. Therefore, we further employed a dual-luciferase assay system to investigate which kind of transcriptional activity that *FvSPL10* possesses (Figure 2B). The results showed that when co-transformation occurs with the reporter vector, the value of LUC/REN ratio with *FvSPL10* is 2.5 folds higher than the negative control (Figure 2C). This indicates that *FvSPL10* possesses transcriptional activation activity. In short, the subcellular localization and transcriptional activity assay results confirmed that *FvSPL10*, similar to its homologs, is a transcriptional activator in plant. 

### 2.3. Interaction of FvSPL10 with the GTAC Motif in FvAP1 Promoter in Vitro

Numerous studies have confirmed that SPL TFs could bind to the GTAC motif [16,28]. For example, the first two *SPL* genes in *A. majus* could be identified because they could bind to the GTAC motif in the promoter of the floral meristem identify gene *SQUAMOSA* [2]. In this study, we observed that in woodland strawberry many flowering-related genes such as *FvAGL24*, *FvSOC1*, *FvTFL* and *FvAP1* all contain at least one GTAC motif in their promoter, among them, there are five GTAC motifs in the promoter of *FvAP1* (Figure 3A). Therefore, we applied electrophoretic mobility shift assay (EMSA) to confirm the interaction between *FvSPL10* and the GTAC motifs in the *FvAP1* promoter. The results indicated that the recombinant *GST-FvSPL10* protein could form DNA-protein complexes when incubated with biotin-labeled probes that contain a fragment of *FvAP1* promoter including one GTAC motif. However, the DNA-protein complexes were gradually abolished with the increasing concentrations of competitors, but not abolished when incubated with mutated probes (Figure 3B). In summary, these results showed that *FvSPL10* could specifically recognize and bind to the GTAC motifs in the promoter of *FvAP1*. 

### 2.4. FvSPL10 Activates the Transcription of FvAP1 in Vivo

As luciferase assay and EMSA have respectively confirmed, *FvSPL10* is a transcriptional activator and *FvSPL10* has binding ability to the promoter of *FvAP1*. We further applied a dual-luciferase transient expression system to confirm whether *FvSPL10* could activate the transcription of *FvAP1* in vivo. As illustrated in Figure 4, when the transient expression effector vector containing *FvSPL10* co-transformed with the reporter vector containing the promoter of *FvAP1* in tobacco leaves, the value of LUC/REN ratio was significantly higher than those of empty effector vector with the reporter (Figure 4). These results confirmed that *FvSPL10* could activate the transcription of *FvAP* in vivo.

### 2.5. Ectopic Expression of FvSPL10 in Arabidopsis Promotes Flowering

In *Arabidopsis*, *AtSPL9* has multiple functions. These functions include controlling vegetative phase change [19], anthocyanin biosynthesis [20] etc. To analyze the functions of *FvSPL10*, we fused *FvSPL10* downstream of the CaMV35S promoter in the binary expression vector pJX001 to generate *FvSPL10* overexpression construct (*FvSPL10-OE*). We then transformed it into *Arabidopsis* through *Agrobacterium*-mediated transformation approach. Nine independent transgenic lines were obtained. Compared with wild-type plants, *FvSPL10-OE* plants significantly promoted flowering under long-day conditions (3–5 days early) (Figure 5A,B). 

In *Arabidopsis*, many *SPL* genes, including *AtSPL9*, can directly target and activate the expression of several *MADS*-box genes [12]. To analyzed whether *FvSPL10* possesses similar functions as *AtSPL9,* we further detected the expression levels of several known *MADS*-box and other flowering-related genes such as *AtFUL*, *AtCO*, *AtAGL42*, *AtLFY* and *AtAP1* by quantitative real-time reverse transcriptase-polymerase chain reaction (qRT-PCR). The results showed that all these flowering-related genes were significantly up-regulated in *FvSPL10-OE* lines, especially *AtAP1* which expressed at least eight folds higher than in the wild-type plants (Figure 5C). 

### 2.6. Ectopic Expression of FvSPL10 in Arabidopsis Enlarges Organs Size

Apparent difference was observed in some organs between *FvSPL10-OE* transgenic lines with wild-type plants. At the seedling stage, *FvSPL10-OE* lines exhibited longer root than the wild-type plants (Figure 6A). At the reproductive stage, the floral organs of *FvSPL10-OE* lines were larger than the wild-type plants (Figure 6B). In addition, *FvSPL10-OEs* produced longer and larger siliques than the wild-type plants (Figure 6C). Finally, we observed that the seeds of *FvSPL10-OEs* were larger than the wild-type plant seeds (Figure 6D,E). These findings indicate that not only does *FvSPL10* regulate plant phase transition, but it also has other phenotypic consequences, such as regulating organ size in *Arabidopsis*. 

## 3. Discussion

*SPLs* are plant-specific TFs that play important roles in plant growth and development. In model plants like *Arabidopsis* and rice, nearly all the members of *SPLs* have been identified and characterized. However, few research studies have been performed on strawberry *SPL* genes. In this study, one *SPL* gene, *FvSPL10*, was cloned from woodland strawberry on the basis of our previous study [8]. Phylogenetic analysis indicated that *FvSPL10* was the ortholog of *AtSPL9* (Figure 1A). Further sequence alignment indicated the two *SPLs* only shared 36% amino acid identity, and the highest similarity region was the SBP domain (Figure 1B). However, to most *SPLs*, the functional regions are the variant parts, not the highly conserved SBP domain that is required for DNA binding ability. For example, in bread wheat, Liu et al. identified that the N-terminal of the TaSPL3/17 was required for interacting with TaD53 (DWARF53) and TOPLESS proteins to regulate the strigolactone signaling pathway [29]. In addition, Yu et al. found the C-terminal of *AtSPL9* was required for physical interaction with DELLA proteins to regulate plant flowering [30]. Considering the low sequence similarity, except the SBP domain between *FvSPL10* and *AtSPL9*, it is reasonable to speculate that although *FvSPL10* might share similar functions to *AtSPL9*, it might have different functions as well.

To explore the biological functions of *FvSPL10*, we first analyzed its subcellular localization and transcriptional activity in tobacco leaves through agroinfiltration-based transient gene expression approach, because it is difficult to carry out the same study in strawberry leaves. As expected, *FvSPL10* was a typical TF that localized in the nucleus with transcriptional activator activity (Figure 2). *SPL* genes were first identified from *A. majus* because they were found binding to the GTAC motif in the promotor of the floral identify gene *SQUAMOSA* [2]. In this study, we also confirmed that *FvSPL10* could recognize and bind to the same motifs in the promotor of *FvAP1* (Figure 3). Further transcriptional activity assay indicated that *FvSPL10* can promote the expression of *FvAP1* (Figure 4). In *Arabidopsis*, *AtSPL9* directly activate flowering-promoting *MADS*-box genes to promote plant phase transition [12]. To verify the functions of *FvSPL10*, we first tried to transform *FvSPL10* into woodland strawberry. However, due to the low transformation efficiency and time-consuming process of strawberry genetic transformation, we have not obtained genetically transformed strawberry plants. We therefore ectopically expressed it in *Arabidopsis*. As expected, compared to the wild-type plants, *FvSPL10-OE* plants exhibited early flowering (3–5 days earlier) under long-day conditions (Figure 5A,B). Several flowering related genes, such as *AtAP1*, *AtFUL*, *AtCO*, *AtAGL42* and *AtLFY* were significantly up-regulated (Figure 5C). More interestingly, apart from leading to early flowering, ectopic expression of *FvSPL10* in *Arabidopsis* resulted in other phenotypic changes, such as elongated roots and enlarged siliques and seeds (Figure 6). It was interesting to observe that there was only partial phenotypic similarity between transgenic *Arabidopsis* that overexpression *AtSPL9* and *FvSPL10*. Except significantly promoted flowering in both transgenic lines, the former was found reduced the content of anthocyanin [20], while the latter was found increased organs size in this study. Interestingly, the phenomena of increasing seed size were observed when two grain-development-related *SPL* genes, *OsSPL16* and *TaSPL16*, were overexpressed in *Arabidopsis* [27,31]. In strawberry, seed-derived auxin plays a critical role in fruit growth and development, and removing the seeds from strawberry fruit resulted fruit growth ceasing [32]. Therefore, in the future, besides investigating the roles of *FvSPL10* in plant phase transition and its relationship to flowering, it is also worth investigating whether *FvSPL10* has functions like *OsSPL16* and *TaSPL16,* which could regulate strawberry seed development and eventually influence fruit development through strawberry transformation. 

## 4. Materials and Methods

### 4.1. Plant Materials and Growth Conditions

Seeds of Woodland strawberry (*Fragaria vesca* accession Hawaii 4) were gifted by Professor Janet P. Slovin (USDA). Plants were grown in the growth chamber with 16 h light at 25 °C and 8 h dark at 20 °C. Plant tissues were collected and immediately frozen in liquid nitrogen then stored at −80 °C for future analysis. *Arabidopsis thaliana* accession Columbia-0 was used as wild-type, plants were grown at 22 °C under long-day conditions (16 h light/8 h dark) in a greenhouse. 

### 4.2. Nucleic Acid Isolation

Genomic DNA was isolated from young leaves of woodland strawberry according to the cetyltrimethylammonium bromide (CTAB) method [33]. Total RNA of woodland strawberry was extracted from runners by using a Plant Total RNA Isolation Kit Plus (Foregene, Chengdu, China) according to the manufacturer’s protocol. First-strand cDNA was synthesized by using PrimeScript™ II 1st Strand cDNA Synthesis Kit (TAKARA, Dalian, China) according to the user manual instructions. 

### 4.3. Bioinformatic Analysis

For phylogenetic analysis, the amino acid sequences of *AtSPLs* and *FvSPL10* were aligned with the Clustal W algorithm and the unrooted radial tree was generated by MEGA 6.0 software using neighbor-joining (NJ) method with 1000 bootstrap replicates [34]. 

### 4.4. Subcellular Localization Analysis of FvSPL10

The cDNA of *FvSPL10* was inserted downstream of the GFP reporter of the binary vector pJX002 (primers are listed in Appendix A). The binary vector pJX002 harboring 35S: GFP-*FvSPL10* expression cassette or the empty pJX002 vector were separately transformed into *A. tumefaciens* strain GV3101 and then infiltrated into tobacco (*Nicotiana benthamiana*) leaves as described by Cheng et al. [35]. Transient expressed *GFP-FvSPL10* or GFP was detected by using a fluorescence microscope (FV3000, Olympus, Tokyo, Japan) two days after infiltration. 

### 4.5. EMSA

The cDNA of *FvSPL10* was cloned into the prokaryotic expression vector pGEX-6P-1 (GE Healthcare) to generate a *GST-FvSPL10* expression construct, which was further transformed into *Escherichia coli* strain BL21 (DE3). The *GST-FvSPL10* fusion protein was induced by 0.3 mM isopropyl thio-β-d-galactoside (IPTG) at 16 °C overnight. The *GST-FvSPL10* fusion protein was lysed from bacteria by sonication and further purified by glutathione-superflow resin. 

A 47 bp fragment of *FvAP1* promoter containing the second GTAC motif was selected as a probe. The probe was labeled by biotin at the 3′ end, and the GTAC motif mutated or non-mutated probes without biotin labeling were used as competitors. Chemiluminescent EMSA Kit (Beyotime Biotechnology, Shanghai, China) was used to perform EMSA following the user manual instructions. The protein-DNA complexes were separated by native polyacrylamide gel electrophoresis and detected by chemiluminescence method. All primers are listed in the Appendix A.

### 4.6. Transcriptional Activity Analysis of FvSPL10

Dual-luciferase assay system was employed to analyze the transcriptional activity of *FvSPL10* in tobacco leaves. The reporter was adopted based on the pGreenII 0800-LUC vector, and the effector construction was adopted based on a CaMV35S promoter-driven pBD vector [36]. The activity was measured as described by Cheng et al. [35], in order to study the trans-activation of *FvSPL10* to *FvAP1*. The promoter of *FvAP1* (1000 bp upstream from the start codon, the nucleotide sequence was listed in Appendix A) was cloned into the pGreenII 0800-LUC vector as reporter, and the *FvSPL10* was inserted into the pEAQ vector as effector. The reporter and effector vectors were transiently expressed in tobacco leaves through *Agrobacterium* infiltration. The LUC/REN activity was detected by using a DLR assay system kit (Promega, Madison, WI, USA) two days after infiltration. The trans-activity of *FvSPL10* to the promoter of *FvAP1* was calculated as described by Cheng et al. [35]. Primers are listed in the Appendix A. 

### 4.7. Construction of FvSPL10 Overexpression Vector and Arabidopsis Transformation

The full-length of *FvSPL10* was cloned into the binary vector pJX001 downstream of the CaMV 35S promoter and transformed into *Arabidopsis* wild-type through *A. tumefaciens*-mediated floral-dip method [37]. Transgenic plants were screened on MS plates supplied with 30 mg/L Hygromycin and further verified by genomic PCR. Plants were grown in greenhouse conditions described above, and T3 generation transgenic lines were used for analysis.

### 4.8. Gene Expression Analysis

QRT-PCR analysis was performed to detect the transcription levels of flowering genes. Total RNA was isolated from 2-weeks old *Arabidopsis* plants by TRIzol reagent (Invitrogen). First-strand cDNA was synthesized by using the Prime Script RT reagent kit according to the manufacturer’s instruction (TAKARA, Dalian, China). 

QRT-PCR was conducted on the CFX96 touch real-time PCR detection system (Bio-Rad, Hercules, CA, USA). The amplification parameters were 95 °C for 2 min, 40 cycles at 94 °C for 5 s then 60 °C for 30 s. The relative expression levels of flowering related genes were calculated using the 2^−ΔΔCT^ method [38], and *Atβ-tubulin2* was used as an internal reference. Three biological replicates and three technical replicates were conducted. All primers used were listed in the Appendix A. 

### 4.9. Phenotypic Measurement and Statistical Analysis

The flowering time of *Arabidopsis* plants was counted from the day of germination to the day of first bud emergence. At least 10 plants were counted for each genotype. Severn-day-old seedlings of wild-type or transgenic lines growing on MS plates were used to measure the root length. For measurement the size of seeds. Dry seeds of the wild-type or transgenic plants were randomly selected and photographed by Stereo microscope (SZ61, Olympus, Tokyo, Japan). The seeds size were imaged by Image J software [39]. At least 30 seeds were measured for each genome type.

## 5. Conclusions

In conclusion, *FvSPL10*, one of the *SPL* genes from woodland strawberry, was cloned and characterized in this study. Transgenic experiments demonstrated that *FvSPL10* has multiple functions in regulating plant growth and development. These findings paved the foundation for further investigation of the function of *FvSPL10* in plant architecture and flowering-related traits of strawberry.

## Figures and Tables

**Figure 1 plants-08-00342-f001:**
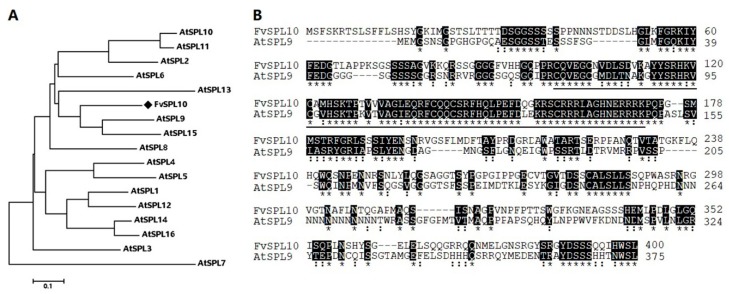
Bioinformatic analysis of *FvSPL10*. (**A**) The amino acid sequences of *FvSP10* and 16 *AtSPLs* were aligned using the Clustal W algorithm, and the neighbor-joining tree was constructed by MEGA6.0 software with 1000 bootstrap replicates. The scale bar represents a distance of 0.1 substitutions per site. *FvSPL10* is marked with a diamond. (**B**) Amino acid sequences alignment of *FvSPL10* and *AtSPL9*. Identical amino acids were highlighted in black and marked by star, and similar amino acids were marked by double dots. In addition, the conserved *SQUAMOSA*-promoter-binding (SBP) domain in their sequences is underlined.

**Figure 2 plants-08-00342-f002:**
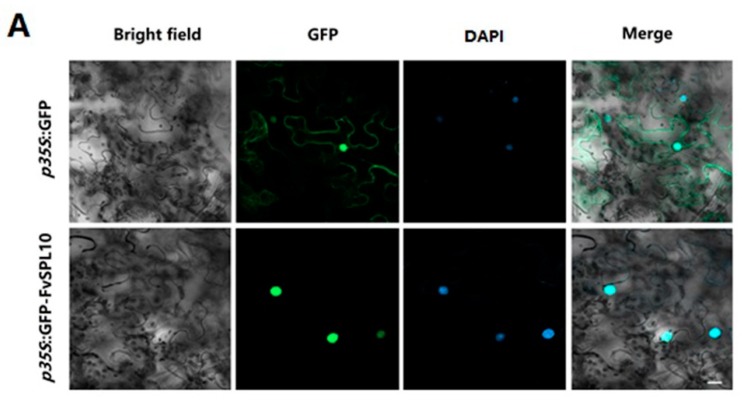
Subcellular localization and transcriptional activity analysis of *FvSPL10*. (**A**) Subcellular localization of *FvSPL10*. Binary vectors containing *GFP-FvSPL10* or *GFP* were transiently expressed in tobacco leaves through the *Agrobacterium* infiltration approach. The fluorescence was detected 48 h after infiltration, and the nuclei were stained by 4′,6-Diamidino-2-phenylindole (DAPI). Bar, 20 μm. (**B**) Schematic representation of the reporter and effector constructs for transcriptional activity analysis. LUC, firefly luciferase; REN, renilla luciferase. (**C**) Transcriptional activation ability of *FvSPL10* in tobacco leaves. Reporter vector was co-transformed into tobacco leaves by *Agrobacterium* with pBD empty vector control or *FvSPL10*, respectively. Each value represents the means of six biological replicates. The ratio of LUC/REN of the pBD control was used as a calibrator (set to 1). Double stars represent a significant difference between the sample and control at *p* < 0.01, based on the Student’s *t*-test.

**Figure 3 plants-08-00342-f003:**
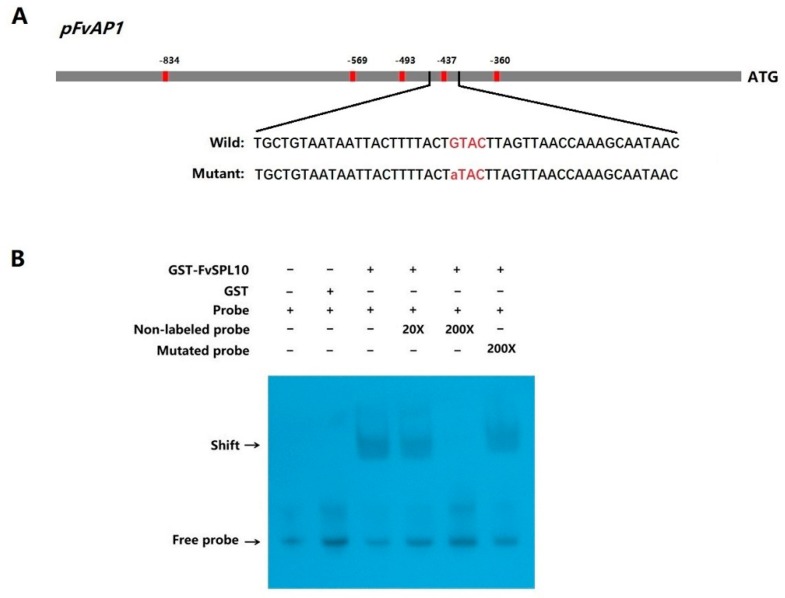
Electrophoretic mobility shift assay (EMSA) verification the binding capability of *FvSPL10*. (**A**) Schematic diagram of the *FvAP1* promoter (−1000 bp upstream from the start codon) and probes used for EMSA. The five red bars represent the five GTAC motifs in the promoter and the number represents the position of these motifs in the promoter. A total of 47 bp oligonucleotides containing the second GTAC motif was selected as the probe. The mutant probe was the same as the wild-type probe except G substituted by A in the GTAC motif. (**B**) Binding of *FvSPL10* to the promoter of *FvAP1* in EMSA. The biotin-labeled probes were incubated with purified *GST* or *GST-FvSPL10* fusion protein to detect the binding ability, and the unlabeled or mutant probes were used as competitors to test the binding specificity of *FvSPL10*. Formed DNA-protein complexes were separated by native polyacrylamide gel. Arrows indicate the position of the shifted bands and free probes, respectively.

**Figure 4 plants-08-00342-f004:**
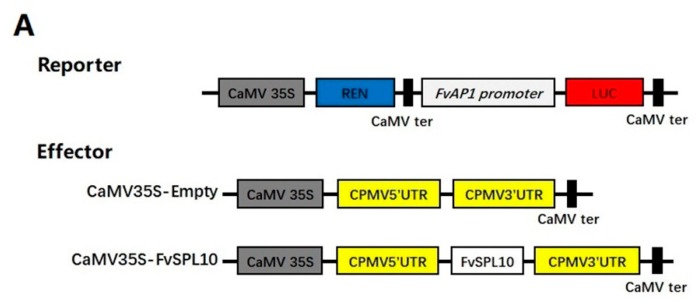
*FvSPL10* directly activates the expression of *FvAP1*. (**A**) Schematic representation of the reporter and effector constructs used in this study. LUC, firefly luciferase; REN, renilla luciferase. (**B**) *FvSPL10* activates the promoter of *FvAP1*. The activation was calculated by the ratio of LUC/REN between sample and control. The ratio of LUC/REN of empty vector together with *FvAP1* promoter was used as a calibrator (set to 1). Each value represents the means of six biological replicates. Double stars represent a significant difference between the sample and control at *p* < 0.01, based on the Student’s *t*-test.

**Figure 5 plants-08-00342-f005:**
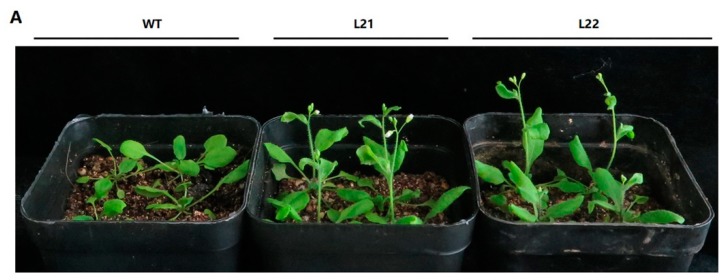
Overexpression of *FvSPL10* in *Arabidopsis* promotes early flowering. (**A**) Phenotypes of 3-week-old plants under long-day conditions. Wild-type (WT) *Arabidopsis*; L21, L22, homozygous transgenic lines with overexpression of *FvSPL10*. (**B**) Comparison of flowering time between WT and transgenic plants. The value of each column represents mean SD (*n* = 12). (**C**) Relative transcript levels of flowering-related genes in WT and transgenic lines. Two transgenic lines, L21 and L22 were selected as representatives. Single or double stars at the top of each column in the figure represent a significant difference among different lines at 0.01 < *p* ≤ 0.05 and *p* < 0.01, respectively, based on the Student’s *t*-test.

**Figure 6 plants-08-00342-f006:**
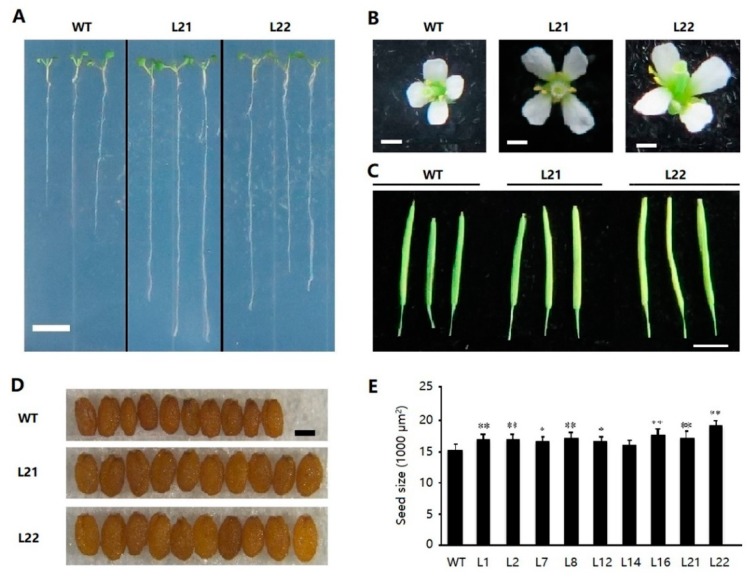
Overexpression of *FvSPL10* in *Arabidopsis* enlarges plant organs. Wild-type (WT) *Arabidopsis*; L21, L22, transgenic lines with overexpression of *FvSPL10*. (**A**) Roots of 7-day-old WT and transgenic plant seedlings. Scale bar, 10 mm. (**B**) Floral organs of WT and transgenic plants. Scale bar, 1 mm. (**C**) Siliques of WT and transgenic plants. Scale bar: 10 mm. (**D**) Mature seeds of WT and transgenic plants. Scale bar: 0.1 mm. (**E**) Statistics of seed size between WT and transgenic plants. Seed size was calculated by ImageJ software. Data are given as mean SD (*n* = 30) by Student’s *t*-test. Single or double stars at the top of each column in the figure indicate a significant difference among genotypes at 0.01< *p* ≤ 0.05 and *p* < 0.01, respectively, based on the Student’s *t*-test.

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
