# Peer review of "Molecular Cloning and Characterization of SQUAMOSA-Promoter Binding Protein-Like Gene FvSPL10 from Woodland Strawberry (Fragaria vesca)"

_plants, 2019, doi:10.3390/plants8090342_

Round 1
Reviewer 1 Report
This study focuses on characterizing the SPL-family transcription factor FvSPL10 from woodland strawberry, which the authors previously found to be mainly expressed in vegetative tissues. To this effect, they expressed a GFP-FvSPL10 transgene in tobacco leaves and showed that it localizes to the nucleus and is active in this system, using a dual-luciferase reporter. Then, they show that recombinant GFP-FvSPL10 can bind to a FvAP1 promoter fragment in vitro, as well as in a dual-luciferase assay in tobacco leaves. Finally, the authors overexpress FvSPL10 in Arabidopsis and found a relatively mild early flowering and organ size phenotype.
The two main shortcomings of the study are that: (1) unfortunately many background questions are not explained. What are the key roles of AtSPL9 that prompted the authors to concentrate on FvSPL10? Why was the FvAP1 promoter in particular selected as a target for EMSA? The authors show two GTAC motifs in this promoter. Are there more GTAC motifs on this promoter? How many genes have a GTAC motif on their promoter? Or was FvAP1 selected in particular because AtAP1 is a flowering-related gene and the effect of overexpressing FvSPL10 in Arabidopsis was early flowering? The authors should make their reasoning more explicit.
(2) The title of the paper is misleading. Indeed, the strawberry FvSPL10 gene was not functionally characterized “in woodland strawberry”, but as transgenes in tobacco infiltrations and overexpressed transgenes in Arabidopsis thaliana. These experiments showed that FvSPL10 is an active transcription factor, but do not give any information as to its endogenous expression domain, nor its endogenous targets in woodland strawberry. It is therefore, in my understanding, too early to say that this transcription factor was “functionally characterized” and that the findings “might be applied in strawberry molecular breeding”.
The authors do not explain whether it is possible or not to conduct leaf infiltration experiment on strawberry leaves, for example. Someone outside of the field of strawberry studies may not know whether this is difficult or not. The same remark can be said of obtaining a strawberry FvSPL10 mutant by CRISPR/Cas9, and/or of overexpressing FvSPL10 directly in strawberry – is this feasible? These experiments would be much more informative than what is currently presented.
Additionally:
– Figure 1A only compares FvSPL10 to Arabidopsis SPL genes. Given that AtSPL9 and AtSPL15 are functionally redundant, it would be interesting to know whether strawberry also possesses a SPL redundant to FvSPL10. The authors should produce a phylogenetic tree that includes not only all Arabidopsis SPL genes, but also all strawberry SPL genes, to give a more accurate picture.
– It is well known that AtSPL9 expression is regulated by miR156 in Arabidopsis. Does woodland strawberry possess this miRNA and/or this miRNA target site on the FvSPL10 gene?
– The authors tested that FvSPL10 can bind to a fragment of the FvAP1 promoter. Did the authors determine Kd and Ki values, which are typically calculated when making EMSA assays? If so, is the Kd value similar to what is expected for AtSPL9?
Moreover, while it is indeed likely that FvSPL10 binds to the GTAC motif, this particular experiment does not prove it: technically, the bound motif could for example be ACTTA, which is mutated in the mutant probe to AATTA.
It should also be stressed that the manuscript will require extensive English editing for style and clarity, as many sentences are very hard to read in the present form (which unfortunately includes the abstract and even the first sentence of the paper).
Reviewer 2 Report
Reviewing manuscript by Xiong et al in Plants.
In this manuscript, the research team characterized the SPL10 of woodland strawberry as a transcription factor that its overexpression enhanced the growth and size accumulation of Arabidopsis plants. The SPL proteins are transcription factors that play important roles in plant growth and development. They were well characterized in model plants and therefore it is important to look for the activity in other agriculture important crops. In this work, the authors performed experiments to see where in the cell the FvSPL10 is localized, how it effects the expression of reporter genes and how ectopic expression in Arabidopsis effected plant growth. The results suggest that the FvSPL10 functions similarly to the homologs from other plants. The work is generally well done and the experiments nicely performed and presented. However, several controls are missing and some issues should be taken care before publication.
1. Fig 1 panel B: The amino acid alignment is not presented clear in the PDF version I see. In addition, it would be nice to show also the similar amino acids. Not only the identical.
2. Line 99: the word “the” is repeated twice.
3. Fig 2A: This experiment need a positive control in which the nucleus is stained. Then the overlap of GFP and nucleus (DNA) staining could be observed. As it is now, it does not say that the protein is located to this organelle.
4. Fig 2C: The promoter sequence of the construct is not clear. Should be presented. Is 2.5 increase is characterized to this factor? Compare to the results in other model plants.
5. What is the difference between Fig 2B, C and Fig 4A, B? Explain clearly.
6. Fig 3: How comes X20 of non-labeled probe did not affect the signal? The stoichiometric of the components should be optimized. As it is now, the result of adding X20 of mutated probe has no meaning since even the non-mutated did not inhibit the bound protein-DNA signal.
7. Lines 130-131: The arrows do not indicate the shifted bands. Only one of them.
8. Fig 5 and the transgenic plants: a negative control, in which only the vector is introduced, is needed for this experiment.
Round 2
Reviewer 1 Report
The revised version of this manuscript by Xiong et al. addresses some of the comments raised in the previous review. In particular, the authors make a good point that it is technically difficult to infiltrate woodland strawberry leaves and to genetically transform woodland strawberry. I think these limitations are important points that actually need to be stated in the paper, and not just in the response to the previous review, as they will make the experimental choices clearer.
I also still think that the title of the paper is misleading by stating “(…) FvSPL10 in woodland strawberry” as it still implies experiments conducted in woodland strawberry. Two more accurate titles could be:
“Molecular cloning and characterization of Squamosa promoter binding protein-like gene FvSPL10 from woodland strawberry (Fragaria vesca)”
or “Molecular cloning and characterization of Squamosa promoter binding protein-like gene FvSPL10 of woodland strawberry (Fragaria vesca)”
line 194: It is imprecise to say that FvSPL10 has a “function” in Arabidopsis, as it is not an Arabidopsis gene; a more accurate formulation would be to say that it has “phenotypic consequences”. Can the authors compare these to the expected phenotype of an overexpressed AtSPL9?
There are still some typos in the manuscript that should be corrected, such as line 27: “DAN”.
Author Response
Comments and Suggestions for Authors
The revised version of this manuscript by Xiong et al. addresses some of the comments raised in the previous review. In particular, the authors make a good point that it is technically difficult to infiltrate woodland strawberry leaves and to genetically transform woodland strawberry. I think these limitations are important points that actually need to be stated in the paper, and not just in the response to the previous review, as they will make the experimental choices clearer.
RESPONSE: We thank the reviewer’s helpful comments and suggestions.
We have stated the reasons in the revised manuscript:
Lines 221-222: “… in tobacco leaves through agroinfiltration-based transient gene expression approach, because it is difficult to carry out the same study in strawberry leaves.”
Lines 229-232: “To verify the functions of FvSPL10, we first tried to transform FvSPL10 into woodland strawberry. However, due to the low transformation efficiency and time-consuming process of strawberry genetic transformation, we have not obtained genetically transformed strawberry plants. We therefore ectopically …”
I also still think that the title of the paper is misleading by stating “(…) FvSPL10 in woodland strawberry” as it still implies experiments conducted in woodland strawberry. Two more accurate titles could be:
“Molecular cloning and characterization of Squamosa promoter binding protein-like gene FvSPL10 from woodland strawberry (Fragaria vesca)”
or “Molecular cloning and characterization of Squamosa promoter binding protein-like gene FvSPL10 of woodland strawberry (Fragaria vesca)”
RESPONSE: We have revised the title of the paper to “Molecular cloning and characterization of Squamosa promoter binding protein-like gene FvSPL10 from woodland strawberry (Fragaria vesca)” following your suggestion.
line 194: It is imprecise to say that FvSPL10 has a “function” in Arabidopsis, as it is not an Arabidopsis gene; a more accurate formulation would be to say that it has “phenotypic consequences”. Can the authors compare these to the expected phenotype of an overexpressed AtSPL9?
RESPONSE: We have revised the “function” into “phenotypic consequences” in the revised version (Line 194).
Additionally, we also added the comparison in the manuscript.
Lines 237-241: “It was interesting to observe that there was only partial phenotypic similarity between transgenic Arabidopsis that overexpression AtSPL9 and FvSPL10. Except significantly promoted flowering in both transgenic lines, the former was found reduced the content of anthocyanin [20], while the latter was found increased organs size in this study.”
There are still some typos in the manuscript that should be corrected, such as line 27: “DAN”.
RESPONSE: We have carefully checked the manuscript. The errors have been revised in the manuscript that been marked in red.
Reviewer 2 Report
The authors revised the ms according to the review.
Author Response
Comments and Suggestions for Authors: The authors revised the ms according to the review.
RESPONSE: We thank the reviewer’s carefully reading our manuscript and helpful comments and suggestions.